# Alpha-Synuclein: Mechanisms of Release and Pathology Progression in Synucleinopathies

**DOI:** 10.3390/cells10020375

**Published:** 2021-02-12

**Authors:** Inês C. Brás, Tiago F. Outeiro

**Affiliations:** 1Center for Biostructural Imaging of Neurodegeneration, Department of Experimental Neurodegeneration, University Medical Center Göttingen, 37075 Göttingen, Germany; ibras@gwdg.de; 2Max Planck Institute for Experimental Medicine, 37075 Göttingen, Germany; 3Faculty of Medical Sciences, Translational and Clinical Research Institute, Newcastle University, Framlington Place, Newcastle Upon Tyne NE2 4HH, UK; 4Scientific Employee with a Honorary Contract at Deutsches Zentrum für Neurodegenerative Erkrankungen (DZNE), 37075 Göttingen, Germany

**Keywords:** Parkinson’s disease, alpha-synuclein, prion-like spreading, cell-to-cell transfer, neurodegeneration

## Abstract

The accumulation of misfolded alpha-synuclein (aSyn) throughout the brain, as Lewy pathology, is a phenomenon central to Parkinson’s disease (PD) pathogenesis. The stereotypical distribution and evolution of the pathology during disease is often attributed to the cell-to-cell transmission of aSyn between interconnected brain regions. The spreading of conformationally distinct aSyn protein assemblies, commonly referred as strains, is thought to result in a variety of clinically and pathologically heterogenous diseases known as synucleinopathies. Although tremendous progress has been made in the field, the mechanisms involved in the transfer of these assemblies between interconnected neural networks and their role in driving PD progression are still unclear. Here, we present an update of the relevant discoveries supporting or challenging the prion-like spreading hypothesis. We also discuss the importance of aSyn strains in pathology progression and the various putative molecular mechanisms involved in cell-to-cell protein release. Understanding the pathways underlying aSyn propagation will contribute to determining the etiology of PD and related synucleinopathies but also assist in the development of new therapeutic strategies.

## 1. Introduction

More than 200 years ago, James Parkinson described some of the clinical symptoms of the disease that, later, was named as Parkinson’s disease (PD) in “Essay on the Shaking Palsy” [1]. Clinically, patients exhibit a progressive deterioration of neurological functions such as cognition and motor function, but also sleep disorders (rapid eye movement (REM) sleep disorder), hyposmia, and autonomic failure [2]. The motor symptoms result from the severe loss of dopaminergic neurons in the substantia nigra pars compacta (SNpc) and the consequent deregulation of basal ganglia activity [3]. PD is one of several synucleinopathies, which are a diverse group of neurodegenerative diseases known for the deposition of misfolded alpha-synuclein (aSyn) in the brain [4,5]. aSyn can accumulate in Lewy bodies (LBs) or Lewy neurites (LNs), in Lewy body diseases, or in glial cytoplasmic inclusions (GCIs), in multiple system atrophy (MSA) [6,7,8]. 

LBs inclusions contain high levels of aSyn phosphorylated at serine 129 (pS129). In addition, it is estimated that 10 to 30% aSyn in LBs is truncated in the N- or C-terminal regions [9,10,11]. In the brain, synaptic dysfunction and neuronal loss are thought to precede the formation of aSyn inclusions. 

Detailed observations of aggregated aSyn in post-mortem brain tissue from patients at different clinical stages of PD form the basis for the hypothesis that Lewy pathology can progress, as disease progresses, through interconnected brain regions [12,13,14]. The putative neurotoxicity exerted by aSyn aggregates, and due to cell-to-cell transfer, it is correlated with increased severity of the clinical symptoms of PD [15]. Although the molecular mechanisms involved in disease progression remain unclear, several studies in human tissue and in animal models are consistent with the cell-to-cell transmission of pathological aSyn [16]. Importantly, recent evidence suggests that transfer of aSyn aggregates with disease-specific conformations, referred to as strains, may partly explain the existence of distinct synucleinopathies (Figure 1) [17,18,19,20]. Similarly to what happens in prion diseases, aSyn strains are thought to template the aggregation of native aSyn into pathological forms, resulting in the spreading and progression of disease pathology [21]. Currently, the molecular mechanisms and factors modulating aSyn aggregation remain obscure, highlighting the need for further studies. 

In this review, we focus on selected studies supporting the prion-like behavior of aSyn and on the molecular mechanisms involved in the spreading of pathology during disease progression. Further clarification of these concepts will assist in the development of new therapeutic strategies aimed at preventing disease progression in synucleinopathies. 

## 2. aSyn: From Function to Neurotoxicity

The synuclein family comprises three small soluble proteins, alpha-, beta- and gamma-synuclein, that bind to phospholipid membranes [22]. aSyn is encoded by the *SNCA* gene and is composed of three distinct domains, which are defined on their amino acid composition: The N-terminal lipid-binding domain, an amyloid-binding central region (NAC), and a C-terminal disordered region [23]. The N-terminal domain is positively charged and contains seven amphipathic repeats containing a conserved KTKEGV hexameric motif, which enables an alpha-helical structure and interactions with membranes [24,25]. The central NAC region is hydrophobic and is mainly involved in fibril formation and aggregation [26]. Lastly, the C-terminal domain is highly acidic and is used for interaction with metals, small molecules, proteins, and other aSyn domains [27].

aSyn is abundant in the brain, although it also exists in erythrocytes and platelets, as well as in other tissues [28]. In the brain, under normal conditions, aSyn is mainly expressed in neuronal cells, located in the pre-synaptic terminal, and possibly bound to membranes of synaptic vesicles [29,30,31]. Although the precise function of aSyn is still a matter of debate, it is thought to play a role in the recycling of synaptic vesicles [32,33]. In particular, aSyn inhibits synaptic vesicle release and disrupts the SNARE complex-mediated lipid membrane fusion [34,35]. A recent study demonstrated that aSyn interacts with VAMP2 to cluster the synaptic vesicle pools, attenuating their recycling [36].

aSyn exists in equilibrium between the unfolded form in the cytosol and an alpha-helical-rich form when bound to membranes [28,37,38]. Intriguingly, under physiological conditions, aSyn was also described to exist as helically folded tetramers that might be more resistant to aggregation, but these findings remain controversial [39]. In disease conditions, it forms beta-sheet-rich amyloid fibrils that accumulate in the brains of patients [38]. In the aggregation process, natively unfolded aSyn monomers are able to self-aggregate into pathological oligomers and, subsequently, into insoluble fibrils (Figure 1). Interestingly, the interaction of aSyn with dopamine results in its oxidation and in the accumulation of aSyn protofibrils, possibly explaining the increased vulnerability of dopaminergic neurons in PD [40]. 

In addition to neuropathological evidence, genetic evidence associates aSyn with the pathogenesis of PD and other synucleinopathies. Point mutations in the gene encoding for aSyn, as well as genomic duplications or triplications, result in familial forms of PD [41,42,43]. Presently, six missense mutations in the *SNCA* gene have been associated with autosomal dominant PD (A53T, A30P, E46K, H50Q, G51D, and A53E) [44,45,46,47,48,49]. The mutations are clustered within the membrane-binding domain, suggesting the contribution of this region to aSyn dysfunction [50,51,52]. 

The specific factors that trigger aSyn aggregation still remain unclear. Mutations, expression levels, clearance efficiency, saturation of membranes, environmental factors, interactions with other amyloidogenic proteins, and/or with intermediary toxic species, truncation, or post-translational modifications are among the myriad of possible factors [23]. 

The pathological consequences of aSyn dyshomeostasis may themselves exacerbate such dyshomeostasis. These include dysregulation of mitochondrial activity, impairment of endoplasmic reticulum (ER)-Golgi and of synaptic vesicle trafficking, disruption of plasma membrane integrity, impairment of protein clearance systems, or impaired immune system and inflammation responses [23,53,54]. 

## 3. The Concept of aSyn Prionoids and Strains

Prions are misfolded and infectious protein assemblies that are capable of transmitting and propagating a disease [55]. Prions arise due to the aberrant folding of endogenous native cellular prion protein (PrP^C^) into an altered form, which is known as scrapie (PrP^Sc^) [56]. A remarkable feature of PrP^Sc^ is its ability to spread from an infected to a healthy cell, causing the self-propagation of the toxic species throughout the brain. Other characteristic properties of prions include the ability to exist with distinct stable conformations, which are commonly referred as strains. Prions are interindividual transmissible and are the cause of transmissible spongiform encephalopathies (TSE) such as Creutzfeldt–Jakob disease (CJD) or fatal familial insomnia in humans, or mad cow disease in bovines [57,58].

Over the last decade, the terms “prionoid” and “prion-like” have been used to describe the self-propagation, through seeding, of disease-related proteins in an analogy with prion disease [59]. In particular, they define the ability of misfolded proteins to recruit physiological proteins of the same type (to seed) and to induce their conversion into a pathological form that propagates from cell-to-cell. However, the use of this terminology has been one of the most controversial topics in the field, since there is currently no evidence demonstrating the direct transmission of neurodegenerative diseases between individuals, contrary to prion diseases.

Stanley Prusiner, who received the Nobel Prize for his work on prion diseases, proposed in 1987 that misfolded proteins associated with other neurodegenerative diseases might have similar properties. The propagation of these proteins would require a permissive host, a suitable environment for replication and transmission, and possibly long incubation times [59]. Recently, growing in vivo and in vitro experimental evidence has shown that templated conversion may not only be characteristic of PrP^Sc^ but also of other disease-related proteins, such as aSyn, tau, or beta-amyloid (Abeta) [57,60,61,62,63]. 

Strikingly, the aggregation of endogenous aSyn can occur through a homotypic (self-seeding) or heterotypic seeding [64,65,66,67,68,69,70,71,72]. The first term is referred to aSyn templating that requires the presence of the hydrophobic NAC region [26,73]. In contrast, heterotypic seeding refers to the involvement of other proteins in the initiation of aSyn aggregation and pathogenesis (such as Abeta, tau, or huntingtin) [74,75,76,77]. 

The first description of protein “strains” came from the observation of distinct clinical phenotypes in animals after infection with PrP^Sc^ [78]. In humans, at least four different PrP^Sc^ strains exist. They present distinct glycosylation patterns and lead to distinct clinical symptoms, anatomical distribution of the pathology, transmission properties, and seeding proficiencies. It is currently established that the strain-specific properties are encoded in the structure of the misfolded proteins, and these are maintained during the continuous transfer in vitro and in vivo [79]. More recently, it has been proposed that distinct structural conformations, or strains, can also be a feature of other disease-associated proteins, thereby explaining the diverse pathological and clinical phenotypes observed in different neurodegenerative diseases. In fact, it was proposed that the heterogeneity of synucleinopathies might be partly attributed to the accumulation of strains with distinct aSyn conformations. In this context, aSyn strains are assemblies that exhibit distinct biochemical, structural, and physical characteristics and are thought to have different seeding and spreading capacities [17,80,81,82]. Interestingly, the pathological form of aSyn has a beta-sheet-rich structure similar to PrP^Sc^. Other similarities include the abnormal folding of endogenous protein into different strains via a template protein, transfer of misfolded proteins between cells, and pathology propagation in the brain [20,83,84]. Intriguingly, the heterogeneity of synucleinopathies and other neurodegenerative diseases is not usually attributed to an alternate hypothesis that arises from the neuropathological examination of the brain: The simultaneous presence of multiple types of pathologies which could, depending on the relative levels, explain the heterogeneity of the multi-morbid old brain [85].

Nevertheless, several recent studies have uncovered apparent conformational differences in aSyn assemblies among different disorders. For instance, MSA strains have shown different seeding potencies and conformations when compared with the strains present in PD brains [19,84,86,87,88]. In particular, aSyn MSA seeds maintain strain characteristics following successive propagation, and they are more resistant to proteolysis and to inactivation with formalin, similar to PrP^Sc^ [89]. Consistently, MSA pathology progresses more rapidly than PD, suggesting that the pathological seeds in MSA are more toxic and spread rapidly throughout the brain [80,81,89,90]. 

The molecular origin for distinct aSyn strains in humans remains largely unknown. Protein conformations, post-translational modifications, local cellular milieu, or even different cell types can form the basis for distinct bioactivities that, at least in part, explain the heterogeneity of neurodegenerative diseases. In the future, it will be important to investigate how the structural characteristics of different aSyn strains can explain the diverse phenotypes in synucleinopathies. 

## 4. Braak Staging and Prion-Like Spreading Hypothesis

In 2003, Braak and co-workers proposed a staging model for categorizing the progression of pathology in PD through neuroanatomically interconnected regions in the brain [13,14]. This model proposes that environmental factors, such as toxins or inflammatory agents, might trigger the formation of LB pathology in the enteric nervous system (ENS), particularly in the gut, or in the olfactory bulb, which would then spread into the brain [91,92,93,94,95], via the vagus nerve in the direction to the substantia nigra pars compacta [96]. More recently, aSyn inclusions were identified in the heart and stomach of a rat model injected with aSyn assemblies into the duodenum. This suggests an anterograde spreading of aSyn pathology (dorsal motor nucleus of the vagus nerve to the stomach), which is followed by a primary retrograde mechanism [97]. These observations indicate the susceptibility of different neuronal populations to aging, demonstrating a unique spatiotemporal distribution of pathology. Another explanation might be the intercellular transfer of unknown pathogens through preferential routes, resulting in the stereotypical progression of pathology.

In 2008, aSyn-positive LBs were observed in grafted fetal mesencephalic dopaminergic neurons that were transplanted in the striatum of PD patients in an effort to alleviate clinical symptoms [98,99]. The phenomenon of Lewy pathology in grafted neurons was interpreted as having been caused by the transfer of aggregated aSyn seeds to the healthy neurons, supporting the hypothesis that PD might be a prion-like disease [69]. 

## 5. Lewy Pathology: More Than Simply One Mechanism or Hypothesis

Several post-mortem observations support the Braak hypothesis. Lewy pathology is observed in the olfactory bulb in the majority of PD cases [100], and reduced olfaction (hyposmia) is an early indicator of PD [101,102,103]. However, hyposmia is not a reliable indicator in some of the genetic forms of the disease [104]. 

Alterations in the brain–gut–microbiota axis, as enteric pathology and gastrointestinal symptoms, have also been documented in several studies [96,105,106,107,108]. These evidences support the hypothesis that PD pathology can spread from the gut to the brain [109]. Recent epidemiological studies also indicate that truncal vagotomy or appendectomy reduces the risk of developing PD, providing support for the gut-to-brain hypothesis [107,110,111,112]. In animal models, the injection of aSyn assemblies in the olfactory bulb results in their spreading to the brainstem [113,114,115]. Interestingly, injection in the gastrointestinal tract resulted in the formation of aSyn inclusions in the brain, supporting that Lewy pathology can spread from gut to brain [93,97,116,117,118,119]. 

However, Braak staging does not explain all clinical cases and abnormal distribution of aSyn pathology [120]. Strikingly, elderly patients with progressive Lewy pathology can lack clinical symptoms [121,122]. In contrast, patients with advanced symptoms and certain genetic forms of PD lack widespread Lewy pathology [123,124,125,126,127,128]. Intriguingly, the presence of only peripheral pathology in the several post-mortem examinations that have been conducted during the past years was never observed, indicating that the spreading of pathology might not be a driver of disease [129,130]. 

Another weakness of Braak staging is the use of the selective vulnerability of neuronal types, which could make them less capable at clearing aSyn aggregates or more prone for generating aggregated species, to explain the pattern of aSyn pathology distribution. Furthermore, the pattern and propagation of aSyn pathology does not always follow neuroanatomic connectivity, suggesting that other mechanisms, besides trans-synaptic spreading, can be involved in the aSyn distribution throughout the brain [65,131,132]. In addition, no LB pathology was observed in a 14-year-old graft transplantation, indicating that the presence or not of LBs in the patient brain grafts can be associated with the graft environment, the time post-grafting, and individual differences between PD patients [133,134]. This raises the possibility that pathology might be initiated by the microenvironment of the PD brain and not through the cell-to-cell transfer of aSyn [135,136,137]. This would also explain why not all PD patients develop Lewy pathology in the ENS [105,129,138]. 

While human studies have suggested that aSyn pathology might be transmissible intra- and inter-cellularly, the exact nature of the endogenous seeds responsible for this process remains unknown [4]. aSyn oligomers, fibrils, ribbons, and pre-formed fibrils are examples of different types of recombinant strains that can be generated using different chemical/biochemical conditions and have shown distinct cell type preference and neurotoxicity (Figure 1) [139,140]. For example, different buffer and salt conditions can enable the formation of either classic fibrils or twisted assemblies as ribbons [17,139]. The effect of aSyn strains in disease propagation and the study of their propagation from host to grafted tissue has been addressed in several studies by the injection of aSyn assemblies in animal models [69,141,142,143]. After injection, these seeds can cross the blood–brain barrier and reach the central nervous system [144]. While fibrils injection in mice causes a loss of dopamine neurons and motor defects, ribbons result in the formation of aSyn inclusions in oligodendrocytes and replicate a pathological marker of MSA. After injection, recombinant aSyn assemblies can imprint their intrinsic structures by conversion of the endogenous monomeric protein [139,145]. Distinct aSyn strains can also be generated by consecutively passaging aSyn fibrils in cells [140]. 

Multiple lines of evidence suggest that oligomeric aSyn species, which are thought to precede the fibrillar aggregates found in LBs, are the culprits for seeding and neuronal degeneration in PD (Figure 1) [146,147]. The assessment of the impact of these oligomeric species in the formation of aSyn inclusions is usually difficult, because oligomers are inherently transient forms and quickly recruit monomeric aSyn to form fibrils [17,148,149]. Therefore, the term “oligomer” is broad and unspecific, constituting a source of unclarity in the field. Interestingly, it has been proposed that the oligomer concentrations that result in toxicity are different from those that efficiently seed the self-amplification [150]. A minor loss of dopaminergic neurons is observed in animal models after the injection of oligomers into the striatum. In contrast, short fibrillar fragments considerably decrease the number of dopaminergic neurons and result in the formation of aSyn inclusions in the cortex and amygdala. Remarkably, short fibril fragments show stronger effects that are attributed to their ability to recruit monomeric aSyn and spread in vivo and hence contribute to the development of PD-like phenotypes [151]. A number of key questions regarding oligomer toxicity and propagation remain to be elucidated. For instance, the role of oligomers in the cell-to-cell propagation across anatomical connected pathways, and the factors that lead to oligomer formation and accumulation. Clarifications of these topics are particularly important for the development of immunotherapy approaches aimed at targeting toxic forms of aSyn.

Interestingly, aSyn assemblies can be originated from the conversion of the endogenous protein, or through the disaggregation of amyloid fibrils by chaperones that produces both monomeric and oligomeric aSyn [152]. In particular, the chaperone HSP110 diminishes the formation of aSyn aggregates in the brain [153], suggesting a mechanism where these oligomers could seed endogenous competent oligomers that could later be propagated from cell-to-cell. Another possibility is the fragmentation of aSyn aggregates by lysosomal proteases and the release of smaller seeding-competent conformers of aSyn. In fact, low pH increases fibril fragmentation, and it might be replicated in endosomes and lysosomes due to their acidic pH [154]. Characterization of the aSyn seeds produced by disaggregases and protein degradation pathways will assist in the development of therapeutic strategies that modulate aSyn levels in the cells.

Full-length, truncated, and cleaved forms of aSyn can exist intra- and extracellularly. Recently, the relevance of aSyn fragments in the extracellular space was shown not only for spreading but also for aggregation and the formation of different strains. These fragments are also able to mediate the aggregation of endogenous full-length aSyn [155]. 

Pathogenic mutations can also facilitate the intercellular transfer and cytotoxicity of aSyn, contributing to early disease onset and to more rapid progression. For example, it was shown that H50Q and A53T mutations significantly increased aSyn secretion. Furthermore, H50Q, G51D, and A53T pre-formed fibrils efficiently seeded in vivo and acutely induced neuroinflammation [156]. These data indicate that pathogenic mutations augment the prion-like spread of aSyn. 

Mutations in the GBA gene, which encodes the lysosomal enzyme glucocerebrosidase (GCase), are an important genetic risk factor for PD. GCase activity is also reduced in sporadic PD brains and with aging. Loss of GCase activity impairs the autophagy lysosomal pathway, resulting in increased aSyn levels. Furthermore, elevated levels of aSyn result in decreased GCase activity, suggesting that GCase deficiency increases the spreading of aSyn pathology and likely contributes to the earlier age of onset and augmented cognitive decline associated with GBA-PD [157].

## 6. Inconsistencies in the aSyn Cell-to-Cell Spreading Model

A wide range of studies is consistent with the prion-like spreading of aSyn. However, there are several points that still need further clarification. One of the main points against this model is the lack of studies demonstrating that aSyn can be transmitted between individuals [158,159]. In traditional prion diseases, transmission occurs between individuals of the species or even across different species. An important and obvious difference between PrP^C^ and aSyn is the transmembrane nature and extracellular location of PrP^C^, while aSyn is predominantly intracellular. 

Another hypothesis is that aSyn transfer occurs through passive release from damaged or dead neurons and not via an active mechanism. The amount of aSyn in the host cell is a key determinant of aSyn pathology generation and spreading, and it remains to be seen whether PD serves as a reservoir of aSyn in a manner similar to what is observed in wild-type or transgenic animals [160]. 

Additionally, there is still little evidence demonstrating that human brain-derived pathological aSyn can spread. If spreading is an important factor in the progression of PD, then evidence needs to be obtained showing the progressive spread of endogenous localized aSyn pathology through connected circuits (comparable to pre-formed seeding models).

Although there are several studies associating aSyn strains with synucleinopathy pathogenesis, the results found in the literature are not always consistent [63]. This may be due to variability in the methodologies, protocols for the preparation of aSyn assemblies [161], genetic background of the animal model, amount of exogenous aSyn assemblies injected in the animal brain, interference with the expression of mouse aSyn, and time post-injection when the samples were collected. Much greater standardization is needed in all these parameters to enable the comparison of the various results.

The ability of human aSyn seeds to induce the formation of inclusions in animal models is another source of controversy. While some studies demonstrated cross-seeding effects between human and mouse aSyn, other studies described the existence of a species barrier. The compatibility between the exogenous aSyn seeds and the endogenous protein has been suggested as a key element of seeding activity in PD models [162]. 

## 7. Mechanisms for Cell-to-Cell Transfer of aSyn

Surprisingly, as it is considered a cytosolic protein, aSyn is present in several human biofluids including saliva, plasma, cerebral spinal fluid (CSF), and red blood cells [163,164,165,166,167]. Major efforts are underway in an attempt to use extracellular aSyn as a biomarker in synucleinopathies, but the correlation between systemic aSyn levels with disease progression remains a matter of debate, in part due of the identification of aSyn assemblies in healthy controls [164,168,169,170]. 

Non-classic secretory pathways have been proposed to be involved in the release of aSyn from cells. These include both passive and active mechanisms (Figure 2). Passive mechanisms include diffusion through the cell membrane and release through compromised cell membranes. Monomeric aSyn, but not higher-order assemblies such as oligomers or aggregates, can diffuse through the cell membrane [171,172]. This process possibly relies on a membrane translocator, since aSyn cannot pass the lipid bilayer [172,173]. It was recently described that aSyn can also be transferred via gap junctions, which are present between adjacent cells [174,175]. Interestingly, endogenous aSyn localized in the cytoplasm remains trapped inside the cell when compared with exogenously added aSyn that can be taken up and released from the cell via diffusion [172]. This suggests that the release of aSyn through compromised cell membranes has a minor effect in the process [176]. 

A recent study identified 14-3-3 proteins as potential regulators of aSyn transmission, proposing that under dysfunction, they may mediate aSyn oligomerization and seeding [177]. Furthermore, the formation of other aSyn assemblies or post-translational modifications may prevent endogenous aSyn from passively escaping the cell. 

A fraction of the cellular aSyn can be actively secreted via non-classical ER/Golgi-independent exocytosis. The folding state-dependent release of aSyn has been shown in several cell types and is most probably correlated with their function in the cell [70,143,173,178,179,180]. This process also occurs under oxidative stress [181], stress conditions [182], or dopamine treatment [183]. Interestingly, the quantity of aSyn released to the cell media is correlated with the intracellular levels [184]. In addition, the susceptibility of different neuronal populations is linked to their endogenous aSyn expression level, establishing that endogenous aSyn levels play a key role in aSyn prion-like seeding [185].

More recently, the misfolding-associated protein secretion pathway (MAPS) has been identified as an unconventional secretion pathway through an ER-dependent process, for preferentially exporting aberrant cytosolic proteins, including aSyn (Figure 2) [186]. The ER-associated deubiquitylase USP19 contains a catalytic domain with a chaperone activity that allows the recruitment of misfolded proteins to the ER surface for deubiquitylation. Then, the deubiquitylated proteins are encapsulated into late endosomes and secreted to the extracellular space [186]. In addition, the HSP70 co-chaperone DNAJC5 was described to play a key role in the secretion of aggregated aSyn assemblies MAPS [187]. 

Another mechanism involved in the release of aSyn assemblies is through their association with exosomes (Figure 2). Exosomes are small vesicles produced from the fusion of multivesicular bodies (MVBs) with the plasma membrane, resulting in their release to the extracellular space [188]. Exosomes are secreted from various cell types, including neurons, astrocytes, and microglia, and they have a regulatory function in synapses and in the intercellular exchange of membrane proteins [189]. Interestingly, vesicular aSyn is more prone to aggregation than cytoplasmic aSyn, and exosomes isolated from the CSF of patients exhibit higher seeding potency compared with controls [173,190]. It has been described that aSyn can be released by exosomes in a calcium-dependent manner. This can be further exacerbated after lysosomal inhibition, lipid peroxidation, or ATG5 knockdown [173,191,192,193,194,195]. Stress conditions increase the translocation of aSyn into vesicles, thus causing its subsequent release to the extracellular space [182]. Furthermore, exosomes released from microglial cells can play an active role in the process of aSyn transmission to neurons. This process is further enhanced by the release of pro-inflammatory cytokines, resulting in protein aggregation and spreading [196].

Tunneling nanotubes (TNTs) represent a novel type of intercellular communication machinery (Figure 2). These membranous structures mediate the communication between neighboring cells and have been implicated in the transfer of pathological aSyn aggregates [178]. However, it remains unclear whether these structures actually mediate the connection of different cell types in vivo.

Passive diffusion [171], conventional endocytosis [69,142,172,197,198], direct penetration through the plasma membrane [199,200,201], and receptor-mediated endocytosis [202,203,204,205] have been proposed as pathways involved in the internalization of aSyn (Figure 2). 

The reasons for aSyn secretion still remain an open question in the context of the prion-like spreading hypothesis. Technical questions relate to the low levels of aSyn and extracellular vesicles that are secreted by cultured cells to the media, and the detection of different types of aSyn species that might be present in the media. The identification of the molecular mechanisms and proteins responsible for the recognition and secretion of aSyn assemblies may, ultimately, support the development of novel approaches to prevent disease progression, but such research is only in its infancy.

## 8. Conclusions

Emerging evidence supports the concept that cell-to-cell transmission and disease-selective strains underlie disease progression and heterogeneity in synucleinopathies. aSyn propagation is coincident with the progression of PD pathology throughout the brain. Our current understanding of this phenomenon is that aSyn pathology spreading may not be the main driving factor in PD. However, it is not yet well understood how aggregated aSyn can transfer from cell-to-cell to induce synaptotoxicity and neurodegeneration. Additionally, it should be determined if endogenous aSyn pathology spreads between cells. Additional studies of the aggregation process are necessary in order to understand the precise mechanism of aSyn propagation in the brain.

Although several studies use in vivo models to address the prion-like properties of exogenous aSyn strains, self-propagation of the endogenous protein remains to be shown. Furthermore, the different mechanisms described to be involved in aSyn cell-to-cell spreading were observed using in vitro models, but whether these mechanisms occur in the brain of PD patients is still unknown. Overall, several studies suggest that aSyn seeds can be transferred through various cell types, inducing the aggregation of the endogenous protein. 

Regardless, the different factors promoting progressive cell-to-cell transfer of aSyn need to be investigated. In the future, it will be important to understand the exact contribution of different aSyn species to the prion-like spreading of PD pathology, by elucidating their transfer and seeding properties, as well as their toxic effects on recipient cells. Clarification of these questions might support the development of novel types of interventions for PD.

Moreover, it remains unclear whether and how familial PD-associated aSyn mutants propagate throughout the brain, as they present distinct aggregation kinetics and different physicochemical properties. Since most studies do not use patient-derived aSyn assemblies, it is uncertain the extent to which experimental models recapitulate the cell-to-cell transmission in PD.

The spreading of aSyn pathology (spatiotemporal distribution, affected cell types, and morphology) in the nervous system is defined by several factors. These factors should differently influence the spread of pathology among strains, thereby causing distinct disease entities. Therefore, it may be necessary to use disease-specific aggregates in experiments in order to identify therapeutic targets that may be unique among these diseases. In prion disorders, approaches targeting PrP^C^ oligomers are being developed after the observation that only oligomers, not monomers, are infectious. However, considering the limited availability of human brain material, it is indispensable to develop new methodologies that enable the production of sufficient amounts of disease-specific aggregates for research.

The development of new therapeutic strategies has been slow and difficult due to the plethora of possible targets that may be tackled in synucleinopathies. This includes aSyn production, aggregation, toxicity, degradation, and spreading. The use of receptor blocking strategies to inhibit aSyn internalization, or of strain-specific antibodies to decrease the levels of extracellular aSyn, may delay the spreading of pathology, but this also needs to be investigated further.

In total, a deeper understanding of the molecular mechanisms underlying aSyn aggregation and intercellular propagation is important for understanding the pathogenesis of PD and related synucleinopathies, to identify new disease targets, and to develop novel therapeutic strategies to halt disease progression.

## Figures and Tables

**Figure 1 cells-10-00375-f001:**
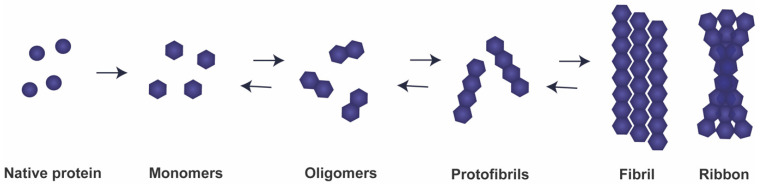
Model for templated misfolding of endogenous alpha-synuclein (aSyn). Under pathological conditions, due to genetic or environmental factors, natively unfolded aSyn monomers are able to self-aggregate in pathological oligomers. These species can be extended into protofibrils and other mature species such as fibrils or ribbons that deposit into inclusion bodies as Lewy bodies (LBs) and Lewy neurites (LNs). Although the biophysical properties and formation of ribbons are still not well understood, the other aSyn assemblies coexist in a dynamic equilibrium and can be transformed into higher- or lower-order conformations.

**Figure 2 cells-10-00375-f002:**
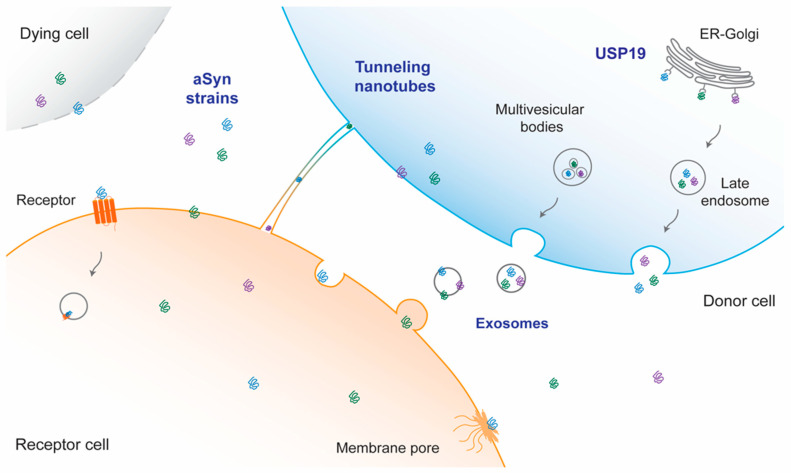
Schematic representation of the possible molecular mechanisms involved in the cell-to-cell transmission and progression of aSyn pathology in Parkinson’s disease (PD). Release of aSyn to the extracellular space can occur via exocytosis/direct translocation through the plasma membrane from a donor to a recipient cell (blue cell). Additionally, misfolded-associated protein secretion pathway (MAPS) is also used to preferentially export aberrant cytosolic proteins. In this mechanism, the endoplasmic reticulum (ER)-associated deubiquitylase USP19 recruits misfolded proteins to the ER surface for deubiquitylation. Then, these cargoes are encapsulated into ER-associated late endosomes and secreted to the extracellular space. Exosomes are derived from multivesicular bodies (MVBs) and have been reported to mediate aSyn release from cells. Tunneling nanotubes (TNTs) can form a direct connection between two cells possibly allowing aSyn from one cell to another. The entry of aSyn into the receptor cell can occur via passive diffusion through the plasma membrane, endocytosis, receptor-mediated endocytosis, and exosome-mediated transfer (orange cell). Furthermore, a high concentration of aSyn in the membrane potentiates its oligomerization and the putative formation of trans-membrane amyloid pores (these pores have yet to be identified in human brain tissue). Last, dying neurons will release their content into the extracellular space, which is another potential source of extracellular aSyn (gray cell).

## Data Availability

Data sharing not applicable.

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
