# Peer review of "Alpha-Synuclein: Mechanisms of Release and Pathology Progression in Synucleinopathies"

_cells, 2021, doi:10.3390/cells10020375_

Round 1

Reviewer 1 Report

The review presents an update of the findings in the mechanisms involved in the spread of diseases associated with alfa-synuclein aggregation and its prion-like behaviour. It is a well-organized manuscript; the only suggestion for the authors is the inclusion of other figures and / or tables to make the paper more appreciable to readers, even those who are not experts in the subject.

Author Response

We thank the reviewer for the constructive comments. Due to the high number of reviews in this field, we decided to focus our manuscript on the current knowledge and controversial issues related with alpha-synuclein spreading in synucleinopathies, instead of reviewing the whole field at length, which tends to be less useful. Furthermore, several recent reviews addressed alpha-synuclein spreading in a more wide-ranging approach with summary tables and figures. We consider that would be repetitive to add more information to this manuscript, as this has been done by others (e.g. as Uemura et al., 2020; Brás et al., 2020; Jaunmuktane et al., 2019; Vargas et al., 2019; Vasili et al., 2019; Burré et al., 2018; Makin 2016). Therefore, we would rather keep it more focused, and we hope the reviewer understands our perspective.

Reviewer 2 Report

The manuscript entitled “Alpha-synuclein: mechanisms of release and pathology progression in synucleinopathies” (cells-1103199) by Drs. Brás & Outeiro provides a relatively comprehensive overview on recent insights into alpha-synuclein based disease mechanisms in Parkinson’s disease and related synucleinopathies.

A large body of literature has been published on the misfolding and aggregation of alpha-synuclein and how the misfolded, aggregated conformer may spread from cell-to-cell and cause the neurodegeneration that is underlying Parkinson’s disease and other synucleinopathies. The similarities to the misfolding and propagation of the infectious prion protein, which causes a variety of diseases in people and livestock, is often seen as a template to explain the molecular and cellular mechanisms of the synucleinopathies. In this review the authors compare and contrast the current knowledge about the prion diseases and synucleinopathies on several occasions.

Overall, the review provides a good overview of alpha-synuclein biology and the related synucleinopathies. However, a few minor items should be addressed to make the review more comprehensive:

  • On page 4 the authors comment on the remarkable cell-to-cell spreading ability of PrPSc. It would be good to add a sentence about the unique individual-to-individual spreading ability of the prions, which is still unmatched among the ‘prionoid’ diseases. Particularly, since this distinction is mentioned later on (e.g. on page 8).
  • On page 5 the molecular origins of the aSyn strains is touched upon. Here it would be fitting to reference the plethora of cryoEM structures that have been published of alpha-synuclein amyloid fibrils (e.g. Guerrero-Ferreira R, Kovacik L, Ni D, Stahlberg H (2020). Curr Opin Neurobiol, 61:89-95).
  • On page 8 the authors briefly mention the many mouse studies that demonstrate ‘neuroinvasion’ of aSyn aggregates when administered peripherally, but only cite one paper. Here it would be beneficial for the general reader to expand this paragraph, add more detail, and include some key citations. The finding that aSyn aggregates can cause pathology in these mouse models upon peripheral injection mirrors studies that suggest similar pathways in synucleinopathy patients, which increases the importance to discuss the similarities to bonafide prion spread.
  • The discussion under the heading "Mechanisms for cell-to-cell transfer of aSyn" would benefit from a paragraph highlighting similar findings for the cell-to-cell spread of aggregated tau protein, which is also an intracellular ‘pathogen’.
  • Finally, at the bottom of page 11 the authors conclude that “it may be necessary to use disease-specific aggregates in experiments in order to identify therapeutic targets that may be unique among these diseases”. A brief (e.g. one sentence) mention of the mechanistic similarities to the prion diseases would be useful.

Author Response

We thank the reviewer for the constructive comments. Due to the high number of reviews in this field, we decided to focus our manuscript on the current knowledge and controversial issues related with alpha-synuclein spreading in synucleinopathies, instead of reviewing the whole field at length, which tends to be less useful. Furthermore, several recent reviews addressed alpha-synuclein spreading in a more wide-ranging approach with summary tables and figures. We consider that would be repetitive to add more information to this manuscript, as this has been done by others (e.g. as Uemura et al., 2020; Brás et al., 2020; Jaunmuktane et al., 2019; Vargas et al., 2019; Vasili et al., 2019; Burré et al., 2018; Makin 2016). Therefore, we would rather keep it more focused, and we hope the reviewer understands our perspective.

Rev2: The manuscript entitled “Alpha-synuclein: mechanisms of release and pathology progression in synucleinopathies” (cells-1103199) by Drs. Brás & Outeiro provides a relatively comprehensive overview on recent insights into alpha-synuclein based disease mechanisms in Parkinson’s disease and related synucleinopathies. A large body of literature has been published on the misfolding and aggregation of alpha-synuclein and how the misfolded, aggregated conformer may spread from cell-to-cell and cause the neurodegeneration that is underlying Parkinson’s disease and other synucleinopathies. The similarities to the misfolding and propagation of the infectious prion protein, which causes a variety of diseases in people and livestock, is often seen as a template to explain the molecular and cellular mechanisms of the synucleinopathies. In this review the authors compare and contrast the current knowledge about the prion diseases and synucleinopathies on several occasions. Overall, the review provides a good overview of alpha-synuclein biology and the related synucleinopathies. However, a few minor items should be addressed to make the review more comprehensive:”

“On page 4 the authors comment on the remarkable cell-to-cell spreading ability of PrPSc. It would be good to add a sentence about the unique individual-to-individual spreading ability of the prions, which is still unmatched among the ‘prionoid’ diseases. Particularly, since this distinction is mentioned later on (e.g. on page 8).”

We thank the reviewer for the positive comments. We added this information on page 4, 3rd paragraph (highlighted with track changes).

“On page 5 the molecular origins of the aSyn strains is touched upon. Here it would be fitting to reference the plethora of cryoEM structures that have been published of alpha-synuclein amyloid fibrils (e.g. Guerrero-Ferreira R, Kovacik L, Ni D, Stahlberg H (2020). Curr Opin Neurobiol, 61:89-95).”

As kindly suggested by the reviewer, we added references to articles proving information about alpha-synuclein strains and cryoEM structures (page 5, 2nd paragraph, highlighted with track changes).

“On page 8 the authors briefly mention the many mouse studies that demonstrate ‘neuroinvasion’ of aSyn aggregates when administered peripherally, but only cite one paper. Here it would be beneficial for the general reader to expand this paragraph, add more detail, and include some key citations. The finding that aSyn aggregates can cause pathology in these mouse models upon peripheral injection mirrors studies that suggest similar pathways in synucleinopathy patients, which increases the importance to discuss the similarities to bonafide prion spread.”

Please notice that in page 8, 2nd paragraph, we are particularly discussing the effect of alpha-synuclein oligomers when compared to fibrils. Other studies regarding aSyn administered peripherally were previously described in the text on page 7, 3rd paragraph. Following the reviewer suggestion, we added further information and citations on page 7 (highlighted with track changes). 

“The discussion under the heading "Mechanisms for cell-to-cell transfer of aSyn" would benefit from a paragraph highlighting similar findings for the cell-to-cell spread of aggregated tau protein, which is also an intracellular ‘pathogen’.”

We agree with the reviewer comment that tau, as alpha-synuclein, is a intracellular ‘pathogen’ and uses similar mechanism to be transferred between cells. However, since recent reviews addressed the similarity of the spreading mechanisms between alpha-synuclein and tau, we felt that would be repetitive to address it in this manuscript (as Uemura et al., 2020; Vasili et al., 2019). Additionally, on page 4 (5th paragraph) we referred to tau and other disease-associated proteins that have been proposed to follow the prion-like spreading hypothesis.

“Finally, at the bottom of page 11 the authors conclude that “it may be necessary to use disease-specific aggregates in experiments in order to identify therapeutic targets that may be unique among these diseases”. A brief (e.g. one sentence) mention of the mechanistic similarities to the prion diseases would be useful.”

As suggested by the reviewer, we made a short reference to a therapeutic strategy being addressed in prion disorders on page 13 (3rd paragraph, highlighted with track changes).

Reviewer 3 Report

In their manuscript “Alpha-synuclein: mechanisms of release and pathology progression in synucleinopathies” the authors provide a thorough overview of the prion-like hypothesis of alpha-synuclein pathologic spread, the implications of different strains of alpha-synuclein, potential inconsistencies in the above hypothesis, and also delineate potential mechanisms of cell to cell transfer of alpha-synuclein.

The points in this review were well- articulated, thoughtfully presented, and well-referenced. There were only a few minor points for comment.

Page 6 paragraph 1; may want to note that impaired olfaction is associated with a majority of Parkinson’s disease, but not a reliable indicator in some genetic forms of Parkinson’s disease.

Page 6, paragraphs 1 and 2 are a bit repetitive (statement that GI symptoms precede motor symptoms) appears in both paragraphs

Page 6 paragraph 3; in addition to the referenced parkin case, also may want to mention that certain genetic forms lack Lewy pathology (ie some LRRK2 mutations have pure nigrostriatal degeneration without Lewy bodies).  

Minor grammatical errors p3 line16 "possible" instead of "possibly", etc.

Overall this review provides a very nice overview of the topic including very recent findings and points out areas where more research is needed to advance the field.

Author Response

“Page 6 paragraph 1; may want to note that impaired olfaction is associated with a majority of Parkinson’s disease, but not a reliable indicator in some genetic forms of Parkinson’s disease.”

We thank the reviewer for the positive comments. We have revised this information in the manuscript (page 6, 4th paragraph, highlighted with track changes). 

“Page 6, paragraphs 1 and 2 are a bit repetitive (statement that GI symptoms precede motor symptoms) appears in both paragraphs.”

We agree and have corrected this information in the manuscript (page 6, 5th paragraph, highlighted with track changes). 

“Page 6 paragraph 3; in addition to the referenced parkin case, also may want to mention that certain genetic forms lack Lewy pathology (ie some LRRK2 mutations have pure nigrostriatal degeneration without Lewy bodies).” 

We thank the reviewer for pointing this out. We have revised the text and added other studies that refer genetic forms of PD that lack Lewy pathology (page 7, 1st paragraph, highlighted with track changes). 

“Minor grammatical errors p3 line16 "possible" instead of "possibly", etc.”

We apologize for this. The error was corrected in the text (page 3, 3rd paragraph, highlighted with track changes).

Reviewer 4 Report

The review manuscript by Brás and Outeiro is fair and comprehensive, written by a group engaging in a research field studying aSyn toxicity for long time. I feel that this review summarizes quite nicely the current understandings and controversial issues need to be addressed in the progression and spread of asyn pathology in PD and related disorders. Below are some points that would help improve the manuscript.

  1. p6 in the sentence starting "However, Braak staging ...", the authors cites #120 in a context mentioning patients lacking Lewy pathology. But the referred paper apparently reports a case with parkin mutation, which might be inappropriate as a counterexample to the sporadic, late-onset cases. Replacing #120 with ones reporting sporadic parkinson cases without Lewy pathology would be better.
  2. Given the ongoing clinical trials testing the therapeutic efficacies of antibodies against aSyn, it would be worth discussing if extracellular aSyn can be targeted by antibodies to cure PD.

Author Response

“p6 in the sentence starting "However, Braak staging ...", the authors cites #120 in a context mentioning patients lacking Lewy pathology. But the referred paper apparently reports a case with parkin mutation, which might be inappropriate as a counterexample to the sporadic, late-onset cases. Replacing #120 with ones reporting sporadic parkinson cases without Lewy pathology would be better.”

We thank the reviewer for the positive comments and for pointing this out. We have revised the text and added references to scientific articles describing sporadic cases. However, we kept the references regarding the genetic cases since these are also important for the discussion of the Braak staging in familial forms of PD (page 7, 1st paragraph, highlighted with track changes).

“Given the ongoing clinical trials testing the therapeutic efficacies of antibodies against aSyn, it would be worth discussing if extracellular aSyn can be targeted by antibodies to cure PD.”

We agree with the reviewer on the importance of recent studies regarding the development of antibodies against extracellular aSyn. However, there are several recent reviews in the literature focusing in aSyn therapies and we consider that would be repetitive to add more information about it (as Vaikath et al., 2019; Zella et al., 2019; Fields et al., 2019). Nevertheless, as suggested by the reviewer, we added a small paragraph addressing this point in the conclusion (page 13, 4th paragraph, highlighted with track changes).